# Association between the Horizontal Gaze Ability and Physical Characteristics of Patients with Dropped Head Syndrome

**DOI:** 10.3390/medicina58040465

**Published:** 2022-03-23

**Authors:** Tatsuya Igawa, Ken Ishii, Ryunosuke Urata, Akifumi Suzuki, Hideto Ui, Kentaro Ideura, Norihiro Isogai, Yutaka Sasao, Haruki Funao

**Affiliations:** 1Department of Orthopaedic Surgery, School of Medicine, International University of Health and Welfare, 852 Hatakeda, Narita City 286-8520, Japan; n.isogai0813@gmail.com (N.I.); sasaospine@marianna-u.ac.jp (Y.S.); hfunao@yahoo.co.jp (H.F.); 2Department of Orthopaedic Surgery, International University of Health and Welfare Mita Hospital, 1-4-3, Mita, Minato-ku, Tokyo 108-8329, Japan; 3Spine and Spinal Cord Center, International University of Health and Welfare Mita Hospital, 1-4-3, Mita, Minato-ku, Tokyo 108-8329, Japan; 4Department of Rehabilitation, International University of Health and Welfare Mita Hospital, 1-4-3, Mita, Minato-ku, Tokyo 108-8329, Japan; ryu.urata62@gmail.com (R.U.); sb11033.1028@gmail.com (A.S.); hideto.3707@gmail.com (H.U.); k.ideura@gmail.com (K.I.); 5Department of Physical Therapy, School of Health Science, International University of Health and Welfare, 2600-1, Kitakanemaru, Ohtawara 323-8501, Japan

**Keywords:** dropped head syndrome (DHS), walking, alignment, horizontal gaze, cervical muscle

## Abstract

*Background and Objectives:* Patients with dropped head syndrome exhibit weakness of the cervical paraspinal muscles. However, the relationship between horizontal gaze disorder and physical function remains unclear. This study aimed to examine and clarify this relationship. *Materials and Methods:* Ninety-six patients with dropped head syndrome were included. We measured the McGregor’s Slope and investigated physical characteristics, including cervical muscle strength, back muscle strength, and walking ability. Factor analysis was used to classify the characteristics of physical function, and a linear multiple regression analysis was used to evaluate independent variables explaining the variance in the McGregor’s Slope. The physical functions of DHS patients were classified into three categories by factor analysis: limb and trunk muscle strength, walking ability, and neck muscle strength. *Results:* The average value of the McGregor’s Slope was 22.2 ± 24.0 degrees. As a result of multiple regression analysis, walking speed (β = −0.46) and apex (β = −0.30) were extracted as significant factors influencing the McGregor’s Slope. *Conclusions:* Horizontal gaze disorders are not associated with cervical muscle strength but with the walking ability and the alignment type of dropped head syndrome.

## 1. Introduction

The main symptom of dropped head syndrome (DHS) is horizontal gaze disorder. A potential cause of impaired horizontal gaze is the severe weakness of the cervical paraspinal muscles. Although neuromuscular diseases including amyotrophic lateral sclerosis and chronic inflammatory neuromuscular disease, Parkinson’s disease, radiation-induced myopathy, and cervical operation cause muscle weakness in the cervical paraspinal muscles, the cause of DHS is still unknown [1]. As a conservative therapy for DHS, patients are often instructed to strengthen the cervical extensor muscles. Endo et al. reported a low therapeutic effect with a 20.9% efficacy rate in conservative therapy for DHS [2]. The low therapeutic effect of conservative therapy may stem from the lack of understanding of the pathophysiology of dropped head (DH). In terms of physical characteristics, we previously reported the reduced trunk muscle mass in patients with DHS [3], suggesting that those with DHS may have a reduced overall/full-body physical function.

Improved clinical symptoms were previously reported in some exercise programs for DHS patients [4,5,6]. However, there are no established exercise programs to date, and there are no physical functions, including muscle strength, that have been identified to explain the improvement in symptoms. Revealing the physical factors associated with the severity of DHS may reveal variables that can be modified by exercise. In addition, categorizing the obtained physical function data can facilitate the clinician’s understanding. These are important information in developing effective conservative treatments. Therefore, the purpose of this study is to clarify the relationship between the horizontal gaze ability and physical characteristics of patients with DHS.

## 2. Materials and Methods

### 2.1. Participants

Patients with DHS who visited our hospital from January 2019 to July 2021 were enrolled in this study. Those over the age of 18 are included. Patients with neuromuscular disease, Parkinson’s disease, radiation-induced myopathy, history of spinal surgery, and refusing to participate were excluded.

### 2.2. Assessment of Physical Function

We investigated the physical characteristics of DHS patients, including the cervical, back, lower limb, and grip strength, in addition to walking ability. Furthermore, we assessed the duration of DH symptoms and neck pain intensity by using a visual analog scale.

#### 2.2.1. Cervical Muscle Strength

The strength of the cervical extensor muscles was measured using the Cervical Extensor Endurance Test (CEET) in a prone position (Figure 1a) [7]. The test was interrupted if the patient opted to discontinue due to neck pain or fatigue, or if trunk extension movements appeared as a compensatory movement. The CEET was completed in 180 s, and if the test was completed once, the second measurement was not performed. The maximum value measured twice was used as the representative value. The Cranio-Cervical Flexion Test (CCFT) was used to assess deep flexion strength in the neck (Figure 1b) [8]. The average value, measured twice, was used as the representative value. In addition, the performance index was calculated as the endurance of the deep cervical flexors [8].

#### 2.2.2. Back Muscle Strength

Back strength was measured using a back strength dynamometer (T.K.K.5402, Takei Scientific Instrument, Niigata, Japan). The average value of the two measurements was corrected by the bodyweight of the subject.

#### 2.2.3. Knee Extensor Strength

Isometric knee extension strength was measured using a lower body strength training machine (leg extension/curl 3530, HUR, Kokkola, Finland) and a performance recorder 9200 (HUR, Kokkola, Finland). Measurements were performed twice on each side, and the average value was corrected by bodyweight.

#### 2.2.4. Grip Strength

Grip strength was measured using a hand grip dynamometer (T.K.K.5401, Takei Scientific Instrument, Niigata, Japan). Measurements were performed twice on each side, and the average value was used as the representative value.

#### 2.2.5. Walking Ability

The time taken to walk 10 m at maximum speed was measured. Measurements were performed twice, and the faster value was used for analysis.

### 2.3. Sagittal Alignment

Horizontal gaze was measured using McGregor’s Slope (McGS), which is the angle of the line from the posterior aspect of the hard palate to the opisthion in relation to a horizontal line [9,10]. McGS was reported to have high reliability and validity for horizontal gaze assessment [10,11]. Sagittal plane alignment was measured using C2–C7 angle (C2C7A), C2–C7 sagittal vertical axis (C2C7SVA), C7–S1 sagittal vertical axis (C7S1SVA), T1 slope, T5–T12 thoracic kyphosis (TK), and apex (cervical or thoracic).

### 2.4. Multiple Imputation

There was a missing value in the measurement data. We planned to use multiple imputations (MI) to account for the missing data values [12]. We created 20 filled-in complete datasets using MI by the chained equation method [13]. Each missing value was replaced with a set of substituted plausible values. In the imputation process, the following covariates were used to create 20 complete datasets: age, gender, height, weight, CEET, CCFT, performance index, back muscle strength, knee extensor strength, grip strength, duration of DH symptoms, and neck pain intensity.

### 2.5. Statistical Analysis

The clinical characteristics and the other measurements were normally distributed according to the Kolmogorov–Smirnov test; therefore, a parametric statistical analysis was performed. Maximum likelihood factor analysis was conducted to clarify the relationship between the physical function data obtained for DHS patients. The number of factors was determined based on satisfying the conditions of the eigenvalue of 1 or more and the factor contribution rate of 60% at the same time, and a factor loading of less than 0.4 was excluded after the varimax rotation. In addition, a simultaneous multiple regression analysis was used to evaluate independent variables explaining the variance in the McGS with gender as the adjustment variable. We considered *p* < 0.05 to be statistically significant in all our analyses, which were performed using SPSS version 27.0 (IBM, Armonk, NY, USA).

## 3. Results

Of the 129 patients who were screened, 96 patients were included in the study. Excluded patients were as follows: Parkinson’s disease (3 patients), radiation-induced myopathy (no patient), history of spinal surgery (8 patients after cervical spine surgery), and refusing (22 patients). Approximately 90% of the DHS patients were female in this study (Table 1). The average BMI was 20.7 kg/m^2^ and the duration of DH symptoms was 22.0 ± 26.3 months. McGS was 22.2 ± 24.0 degrees, C27A was −12.8 ± 27.8 degrees, C27SVA was 60.6 ± 19.9 mm, and TK was 40.6 ± 15.9 degrees. Apex showed a slightly higher proportion of thoracic patients compared to cervical patients.

Physical function parameters had some random missing values (Table 2). The average CEET for cervical extensor strength was 56.1 ± 54.8 sec, and the CCFT for deep cervical flexor strength was 32.7 ± 8.8 mmHg. The back muscle and knee extensor strength were 0.76 ± 0.23 Nm/kg and 1.35 ± 0.54 Nm/kg, respectively. The maximum walking speed was 1.32 ± 0.41 m/sec, and 13 patients (13.5%) walked using any of the assisted-walking devices.

As a result of factor analysis, three factors emerged. These were classified as follows: the first factor included back muscle strength, knee extensor strength, and grip strength; the second factor included walking speed and presence or absence of walking assistance device; the third factor included CCFT and CEET (Table 3). F1, F2 and F3 can explain the severity of DHS in 28.9%, 15.3%, 13.4%, respectively.

There was a weak positive correlation (correlation coefficient 0.23) between the second and third factors. Simultaneous multiple regression analysis revealed that walking speed and apex were significantly associated with McGS (*p*-value < 0.01, adjusted R-squared = 0.212) (Table 4).) Both analyses (using multiple imputed data and complete case data) showed similar results (Table 5).

## 4. Discussion

This study clarified that McGS, which represents horizontal gaze ability, was not significantly associated with muscle strength such as cervical muscle strength, back strength, and lower limb muscle strength, but was rather associated with walking ability and apex. From the results of factor analysis, it was found that the first factor, which integrates the muscle strength of the limbs and the back muscles, knee extensors, and grip strength, is classified into a different category from the cervical muscle strength included in the third factor. The walking ability extracted as the second factor was classified into a category different from the neck, limbs, and trunk muscle strength.

The details of the physical characteristics that provide clues for creating an exercise program for DHS patients have not yet been clarified. Since the cervical spine has a larger range of flexion anatomically than the thoracic spine [14], it can be inferred that the cervical spine-type, with large cervical kyphosis, has significantly larger DH symptoms than the thoracic spine type. McGS is measured at the angle of the line from the posterior aspect of the hard palate to the opisthion. Patients with the cervical type have a more cranial kyphosis of the spine than patients with the thoracic type, and fewer joints to compensate for. A healthy person can walk stably in the upper body by rhythmically contracting the trunk muscles and neck muscles [15]. Control of head movement plays an important role in the attenuation of mechanical perturbations during walking [16]. Since the drooping head is associated with a decrease in walking speed, it can be interpreted as one of the causes of walking instability. The results of this study showed that gait ability was associated with DH, which supports previous studies by Suzuki et al. and Igawa, et al. that investigated the impaired gait ability of DHS patients [17,18]. In the factor analysis, the walking ability was classified as a second factor different from the muscle strength of the neck and limb trunk muscles; thus, it can be interpreted as a separate factor from muscle strength. The results of this study, that the patient’s head droop was associated with severity and walking ability, are very interesting. Although no causal relationship can be drawn from this study, considering previous reports where exercise interventions incorporated gait [17,19,20], we infer that not only static alignment but also dynamic motion represented by walking should be carefully evaluated. There are other reasons why patients with severe DHS have diminished gait. DHS patients who have difficulty in horizontal gaze have their visual field corrected downwards. Visual field loss affects gait performance, such as speed, step length, and stance time [21,22]. Spinal deformities were also reported to reduce gait performance, including gait speed [23,24,25]. The dropped head is also involved in dynamic balance ability. It was reported that in an unbalanced person, the distance between the center of mass and the center of pressure is shortened when walking [26]. Moreover, previous reports have shown that this distance is short and that the balance ability is impaired in DHS patients [18]. The reduced walking speed of DHS patients revealed in this study may reflect decreased balance ability rather than limb and trunk muscle strength.

Furthermore, the result that no significant relationship was found between cervical strength, including extension and flexion, and dropped head is a very surprising finding. Fatigue and damage to the paraspinal muscles were recently shown to cause DH [27], and degeneration of the paraspinal muscles and ligaments were shown to occur in the neck of DHS patients [28]. However, contrary to our hypothesis, DH was not associated with muscle strength in this study. There are three possible reasons. Firstly, the alignment of the cervical spine with other spinal columns results in compensatory adaptive changes [29,30]. Igawa et al. [18] showed that compensatory changes in the thorax also appear during walking in DHS patients. Since the thoracic angles were not unified in the McGS measurements in this study, it is possible that there was no significant association between cervical muscle strength and McGS. Secondly, the limitation of the range of motion of the cervical spine was not taken into consideration. Range of motion affects McGS. It is possible that some cases were unable to raise the head despite good neck muscle strength. Finally, it is possible that integrated muscle strength is involved rather than cervical strength. Previous studies have reported that not only cervical paraspinal muscles but also systemic and gait-specific exercises improved DH [17,19,31].

This study has some limitations. First, this study targeted patients recruited at a single institution. The results of this study are limited due to the possibility of selection bias, and joint research with other institutions is required in the future. Second, since this study is a cross-sectional study, the direct causal relationship is unknown and requires careful interpretation of the results. We do not have control group data to compare with DHS patients. We do not know anything about the strength of muscle and gait parameters in a healthy population. It is unknown whether the physical features of DHS patients observed in the study were indeed abnormal. It is unlikely effective intervention can be developed based on the study results. Finally, the model in this study may have unmeasured data. However, even though DHS is a rare disease, we believe that the data from this study, which analyzed approximately 100 DHS patients, will be very useful information in the future. Conservative therapies for DHS often focus on strengthening the cervical extensor muscles, but the physical functions that affect DH have been unclear. Evaluating the gait ability of DHS patients would provide useful information for clinicians.

## 5. Conclusions

A detailed analysis of the physical function of DHS patients showed that horizontal gaze deficits were not associated with cervical muscle strength but with the walking ability and the alignment type of DHS.

## Figures and Tables

**Figure 1 medicina-58-00465-f001:**
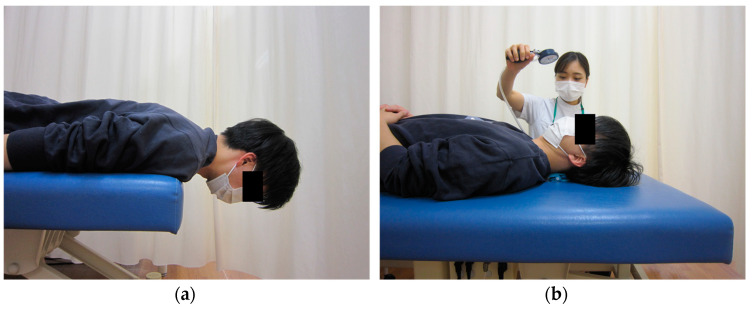
Evaluation of cervical muscle strength. (**a**) CEET and (**b**) CCFT.

**Table 1 medicina-58-00465-t001:** Demographic data of DHS patients.

Variables (*n* = 96)	Value	No. of Missing Data
Age, mean (SD), years	76.7 (9.3)	0
Sex		0
Male, n (%)	11 (11.5)	
Female, n (%)	85 (88.5)	
BMI, mean (SD), kg/m^2^	20.7 (2.3)	0
Duration of DH symptom, months	22.2 (26.3)	0
Radiographic data		
McGS, mean (SD), degree	22.2 (24.0)	0
C2C7A, mean (SD), degree	−12.8 (27.8)	0
C2C7SVA, mean (SD), mm	60.6 (19.9)	0
C7S1SVA, mean (SD), mm	−8.8 (47.4)	0
T1 slope, mean (SD), degree	34.2 (15.5)	0
TK, mean (SD), degree	40.6 (15.9)	0
Apex		0
Cervical, n (%)	40 (41.7)	
Thoracic, n (%)	56 (58.3)	

**Table 2 medicina-58-00465-t002:** Physical function data of DHS patients.

Variables	Value	No. of Missing Data
VAS of neck pain, mean (SD), mm	51.4 (24.3)	6
CEET, mean (SD), sec	56.1 (54.8)	17
CCFT, mean (SD), mmHg	32.7 (8.8)	6
Performance index, mean (SD)	60.3 (31.6)	10
Back muscle strength, mean (SD), kg/kg	0.76 (0.23)	13
Knee extensor strength, mean (SD), Nm/kg	1.35 (0.54)	6
Grip strength, mean (SD), kg	18.3 (6.0)	3
Walking speed, mean (SD), m/sec	1.32 (0.41)	0
Assisted-walking device		0
No used, n (%)	83 (86.5)	
Cane, n (%)	8 (8.3)	
Walker, n (%)	4 (4.2)	
Other, n (%)	1 (1.0)	

**Table 3 medicina-58-00465-t003:** Results of factor analysis.

	Factor Loading		
	F1	F2	F3
Back strength	0.957	0.101	−0.106
Knee extension strength	0.734	0.121	0.331
Grip strength	0.572	0.157	0.267
Walking speed	0.411	0.797	0.441
Presence or absence of walking assistance device	−0.050	−0.567	0.046
CCFT	0.266	−0.185	0.579
CEET	0.012	0.178	0.504
Contribution rate	28.92%	15.31%	13.98%
Factor Correlation Matrix			
F1	1.000	0.045	−0.019
F2		1.000	0.233 *
F3			1.000

The numbers in squares indicate that the factor loading exceeds 0.5; *, *p*-value < 0.05.

**Table 4 medicina-58-00465-t004:** Simultaneous multiple linear regression analysis with McGS as the dependent variable with multiple imputed data.

	Unstandardized Coefficients		Standardized Coefficients	95% Confidence Interval		VIF	*p*-Value
	B	Std. Error	Beta	Lower	Upper		
Age	−0.346	0.287	−0.140	−0.371	0.091	1.441	0.232
BMI	0.206	1.148	0.020	−0.202	0.242	1.351	0.858
Duration of symptom	−0.042	0.094	−0.047	−0.256	0.161	1.132	0.652
Neck pain intensity	−0.005	0.109	−0.005	−0.221	0.212	1.349	0.966
CEET	0.036	0.050	0.082	−0.146	0.310	1.369	0.473
CCFT	−0.306	0.386	−0.103	−0.362	0.156	1.781	0.430
Performance index	−0.025	0.106	−0.033	−0.313	0.247	2.108	0.813
Back strength	−10.872	17.786	−0.095	−0.405	0.215	2.554	0.543
Knee extension strength	10.893	7.298	0.239	−0.080	0.558	2.734	0.140
Grip strength	−0.824	0.918	−0.134	−0.432	0.164	2.375	0.372
Walking speed	−30.064	9.388	−0.481	−0.780	−0.182	2.404	0.002 *
Apex (Cervical/Thoracic)	−12.847	5.025	−0.269	−0.479	−0.059	1.179	0.013 *

F = 2.879; *p*-value < 0.01; adjusted R-squared = 0.212; VIF, variance inflation factor *, *p*-value < 0.05.

**Table 5 medicina-58-00465-t005:** Simultaneous multiple linear regression analysis with McGS as the dependent variable with complete case data.

	Unstandardized Coefficients		Standardized Coefficients	95% Confidence Interval		VIF	*p*-Value
	B	Std. Error	Beta	Lower	Upper		
Age	−0.403	0.436	−0.133	−0.423	0.157	1.455	0.361
BMI	−1.042	1.151	−0.125	−0.404	0.154	1.342	0.370
Duration of symptom	0.017	0.089	0.024	−0.230	0.278	1.098	0.849
Neck pain intensity	−0.077	0.110	−0.096	−0.373	0.180	1.314	0.485
CEET	0.047	0.054	0.129	−0.170	0.425	1.555	0.391
CCFT	−0.123	0.404	−0.049	−0.374	0.276	1.785	0.762
Performance index	−0.171	0.111	−0.242	−0.559	0.074	1.722	0.130
Back strength	−6.923	17.882	−0.074	−0.460	0.312	2.558	0.701
Knee extension strength	10.234	7.103	0.259	−0.104	0.622	2.260	0.157
Grip strength	−0.644	1.011	−0.113	−0.471	0.245	2.207	0.528
Walking speed	−28.049	11.492	−0.456	−0.833	−0.079	2.449	0.019 *
Apex (Cervical/Thoracic)	−12.126	5.459	−0.297	−0.567	−0.027	1.252	0.032 *

F = 2.346; *p*-value < 0.01; adjusted R-squared = 0.230; VIF, variance inflation factor; *, *p*-value < 0.05.

## Data Availability

Not applicable.

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
