# Peer review of "Association between the Horizontal Gaze Ability and Physical Characteristics of Patients with Dropped Head Syndrome"

_medicina, 2022, doi:10.3390/medicina58040465_

Round 1

Reviewer 1 Report

Igawa and colleagues presented a study to examine the association between the horizontal gaze ability and physical characteristics of patients with dropped head syndrome. While this is an interesting topic to clinicians, there are some flaws in the research methods and manuscript.

  1. It is unclear what measure was used to examine horizontal gaze stability in the introduction. This is the main outcome of the study but the authors did not mention how this was measured in the literature and the validity/reliability of the measures.
  2. A lack of healthy controls in the study. It is unclear whether the physical characteristics measured in the patients were within normal limits.
  3.  The inclusion criteria of the patients with dropped head syndrome need to be reported.
  4. While the author used multiple imputation for missing data, it is unclear whether the data were missing completely random. Further, the authors did not provide details about the percentage of missing data in each outcome measure.
  5. More details are needed in statistical analysis. The purpose of performing maximum likelihood factor analysis was not explained.
  6. The Conclusion needs more rework. For example, it is unknown whether the physical features observed in the study were indeed abnormal. It is unlikely effective intervention can be developed based on the study results. 

Author Response

We would like to thank the Reviewer for providing us with detailed comments and an opportunity to revise our manuscript. We have addressed all concerns and have outlined the changes made in the text below.

Reviewer 1

Igawa and colleagues presented a study to examine the association between the horizontal gaze ability and physical characteristics of patients with dropped head syndrome. While this is an interesting topic to clinicians, there are some flaws in the research methods and manuscript.

Comment 1.       It is unclear what measure was used to examine horizontal gaze stability in the introduction. This is the main outcome of the study but the authors did not mention how this was measured in the literature and the validity/reliability of the measures.

Response 1.        We would like to thank Reviewer 1 for providing us with insightful comments. We have assessed the horizontal gaze stability using McGregor's slope. We have included descriptions of the measurement method, reliability, and validity of this measure in our manuscript.

Revised manuscript (page 3, lines 102-105):

Horizontal gaze was measured using McGregor's Slope (McGS) […].

Comment 2.       A lack of healthy controls in the study. It is unclear whether the physical characteristics measured in the patients were within normal limits.

す。

Comment 6.       The Conclusion needs more rework. For example, it is unknown whether the physical features observed in the study were indeed abnormal. It is unlikely effective intervention can be developed based on the study results.

Response 2 and 6.            We wish to express our appreciation to Reviewer 1 for these insightful comments, which have helped us significantly improve the paper. We completely agree with Reviewer`s comment that it is unclear whether the physical characteristics measured in the patients were within normal limits; therefore, this was described as a limitation of the study.

Revised manuscript (page 7, lines 207-211):

We do not have control group data to compare with DHS patients. we don't know […].

Revised manuscript (page 7, lines 215-216):

Evaluating gait ability of DHS patients would provide useful information for clinicians.

Comment 3.       The inclusion criteria of the patients with dropped head syndrome need to be reported.

Response 3.        We would like to thank Reviewer 1 for pointing this out. We have modified eligibility criteria in our manuscript.

Revised manuscript (page 2, lines 58- 59):

Patients with DHS visited our hospital from January 2019 to July 2021 were enrolled in this study. Those over the age of 18 are included.

Comment 4.       While the author used multiple imputation for missing data, it is unclear whether the data were missing completely random. Further, the authors did not provide details about the percentage of missing data in each outcome measure.

Response 4.        We would like to thank Reviewer 1 for this insightful comment. Missing values were randomly occurred in all data. We have added the number of missing data for each outcome measure (Table 1 and 2). Furthermore, both analyses (using multiple imputed data and complete case data) showed similar results (Table 5).

Revised manuscript (page 4, line 134):

Physical function parameters had some random missing values (Table 2).

Revised manuscript (page 4, lines 146-147):

Both analyses (using multiple imputed data and complete case data) showed similar results (Table 5).

Comment 5.       More details are needed in statistical analysis. The purpose of performing maximum likelihood factor analysis was not explained.

Response 5.        We would like to thank Reviewer 1 for advice on describing the statistical analysis. We have included the purpose of conducting a maximum likelihood factor analysis in Introduction and Materials and Methods section.

Revised manuscript (page 2, lines 51- 53):

In addition, the obtained physical function data can be […].

Revised manuscript (page 3, lines 117- 118):

Maximum likelihood factor analysis was conducted to clarify […].

Reviewer 2 Report

Dear authors,

the present study is well designed and presented.

The introduction is synthetically presenting the pertinent background, the methods are well presented. The results section could be improved when implementing the data reporting in a discursive way while avoiding evaluative or discursive statements, which should be reserved for the discussion section.  

Author Response

We would like to thank the Reviewer for providing us with detailed comments and an opportunity to revise our manuscript. We have addressed all concerns and have outlined the changes made in the text below.

Reviewer 2

the present study is well designed and presented.

We would like to thank Reviewer 2 for this insightful comment.

Comment 7.   The introduction is synthetically presenting the pertinent background, the methods are well presented. The results section could be improved when implementing the data reporting in a discursive way while avoiding evaluative or discursive statements, which should be reserved for the discussion section. 

Response 7.    Missing values were randomly occurred in all data. We have added the number of missing data for each outcome measure (Table 1 and 2). Furthermore, both analyses (using multiple imputed data and complete case data) showed similar results (Table 4 and 5).

Revised manuscript (page 4, line 134):

Physical function parameters had some random missing values (Table 2).

Revised manuscript (page 4, lines 146-147):

Both analyses (using multiple imputed data and complete case data) showed similar results (Table 5).

Reviewer 3 Report

The lack of a control group is the main limitation of this paper. 

A Control group is not necessary for a cross-sectional study, but we don't know anything about the strength of muscle and gait parameters in a healthy population.

Author Response

We would like to thank the Reviewer for providing us with detailed comments and an opportunity to revise our manuscript. We have addressed all concerns and have outlined the changes made in the text below.

Reviewer 3

The lack of a control group is the main limitation of this paper. 

We would like to thank Reviewer 3 for this insightful comment.

Comment 1.       The lack of a control group is the main limitation of this paper. A Control group is not necessary for a cross-sectional study, but we don't know anything about the strength of muscle and gait parameters in a healthy population.

Response 1.       

We wish to express our appreciation to Reviewer 3 for these insightful comments, which have helped us significantly improve the paper. We agree with Reviewer 3 that we don't know anything about the strength of muscle and gait parameters in a healthy population.; therefore, this was described as a limitation of the study.

Revised manuscript (page 7, lines 207-211):

We do not have control group data to compare with DHS patients. we don't know […].

Revised manuscript (page 7, lines 215-216):

Evaluating gait ability of DHS patients would provide useful information for clinicians.